# The Evolutionary Young Actin Nucleator Cobl Is Important for Proper Amelogenesis

**DOI:** 10.3390/cells14050359

**Published:** 2025-02-28

**Authors:** Hannes Janitzek, Jule González Delgado, Natja Haag, Eric Seemann, Sandor Nietzsche, Bernd Sigusch, Britta Qualmann, Michael Manfred Kessels

**Affiliations:** 1Institute of Biochemistry I, Jena University Hospital—Friedrich Schiller University Jena, Nonnenplan 2-4, 07743 Jena, Germany; hannes.janitzek@web.de (H.J.); jule.gonzalez@med.uni-jena.de (J.G.D.); nhaag@ukaachen.de (N.H.); eric.seemann@med.uni-jena.de (E.S.); 2Center for Electron Microscopy, Jena University Hospital—Friedrich Schiller University Jena, Ziegelmühlenweg 1, 07743 Jena, Germany; sandor.nietzsche@med.uni-jena.de; 3Department of Conservative Dentistry and Periodontology, Jena University Hospital—Friedrich Schiller University Jena, An der alten Post 4, 07743 Jena, Germany

**Keywords:** cordon-bleu, actin nucleation, ameloblasts, enamel composition, *Cobl* KO (knockout), cortical actin cytoskeleton

## Abstract

The actin cytoskeleton plays an important role in morphological changes of ameloblasts during the formation of enamel, which is indispensable for teeth to withstand wear, fracture and caries progression. This study reveals that the actin nucleator Cobl is expressed in ameloblasts of mandibular molars during amelogenesis. Cobl expression was particularly pronounced during the secretory phase of the enamel-forming cells. Cobl colocalized with actin filaments at the cell cortex. Importantly, our analyses show an influence of Cobl on both ameloblast morphology and cytoskeletal organization as well as on enamel composition. At P0, *Cobl* knock-out causes an increased height of ameloblasts and an increased F-actin content at the apical membrane. During the maturation phase, the F-actin density at the apical membrane was instead significantly reduced when compared to WT mice. At the same time, Cobl-deficient mice showed an increased carbon content of the enamel and an increased enamel surface of mandibular molars. These findings demonstrate a decisive influence of the actin nucleator Cobl on the actin cytoskeleton and the morphology of ameloblasts during amelogenesis. Our work thus expands the understanding of the regulation of the actin cytoskeleton during amelogenesis and helps to further elucidate the complex processes of enamel formation during tooth development.

## 1. Introduction

Formation of enamel, enabling teeth to withstand wear, fracture and caries progression, is ensured by ameloblasts, specialized epithelial cells. Amelogenesis requires ameloblast differentiation, a process that can be divided into four different stages defined by the morphology of the ameloblasts, the pre-secretory phase, the secretory phase, the transformation phase and the maturation stage. In the pre-secretory phase, the cells of the inner enamel epithelium differentiate into preameloblasts by forming cytoplasmic protrusions. These penetrate the basal lamina and lead to the formation of ameloblasts, thereby initiating the secretory phase [1]. The enamel-forming cells transform into tall columnar cells and form Tomes’ processes, conical structures pointing towards the enamel matrix [2]. Ameloblasts secrete large amounts of enamel matrix proteins and this moves them away from the dentin surface [3]. Once the final enamel thickness is achieved, the secretory ameloblasts undergo a retraction of the Tomes’ processes [1]. During this transition, the ameloblasts shorten and adopt a cubic shape. This final stage of amelogenesis is marked by enamel maturation. The enamel-forming cells now promote enamel crystallization by transporting ions into the enamel matrix and secreting kallikrein-4 to remove the matrix proteins [2,3].

In the context of amelogenesis, actin is thought to play a role in the maintenance and plasticity of ameloblast morphology as well as in the movement of enamel-forming cells in the ameloblast layer, which is crucial for the processes of enamel secretion and maturation as well as intertwining to form enamel prisms [4]. Actin– filaments are enriched in the ameloblasts proximally and distally in form of a proximal terminal web (PTW) and a distal terminal web (DTW), where they attach intracellularly to epithelial junctional complexes [5,6].

Actin filament formation requires the action of actin nucleators [7,8]. Among the different actin nucleators identified, Cobl (also named cordon-bleu) [9] exhibits some remarkable, in part even unique features. In contrast to, for example, the Arp2/3 complex, which represents the major actin nucleator for many basic functions of mammalian cells and already has important functions in lower eukaryotes, such as yeast [10], Cobl is an evolutionary relatively young actin nucleator with rather specialized functions in distinct sets of specialized cells that are only beginning to emerge. Cobl is particularly important for the cellular morphogenesis of hippocampal neurons [9,11], for the postnatal development of cerebellar neurons in mice [12] and additionally for post-stroke dendritic arbor regrowth in the cerebral cortex of mice [13]. In embryonic rat neurons, Cobl was observed to specifically concentrate together with F-actin at nascent dendritic branch initiation sites [14,15]. Further specialized cell types relying on Cobl functions include epithelial enterocytes in the small intestine [16] and outer hair cells in the cochlea during the early postnatal stages of mice [17]. In both of these cell types, Cobl has been shown to be important for specialized sets of cytoskeletal structures [16,17].

In addition to representing a rather powerful actin nucleator that can even work with very low levels of ATP-loaded G-actin [9], Cobl was furthermore uncovered to be spatially and temporally regulated by two mechanisms that currently seem rather unique among actin nucleators, posttranslational modification by the protein arginine methyltransferase PRMT2 [18] and furthermore direct modulations of functionally critical Cobl interactions, such as that with G-actin, by calcium-activated calmodulin [14,15].

Analyzing *Cobl* KO mice [17], we here unravel that the actin nucleator Cobl exerts a decisive influence on the actin cytoskeleton and the morphology of ameloblasts during amelogenesis and thereby shed light on a critical molecular effector in this important process.

## 2. Materials and Methods

### 2.1. Mice

Experiments were performed with WT and *Cobl* KO mice. *Cobl* KO mice were obtained by excision of exon 11 of the *Cobl* gene using the Cre/loxP-mediated recombination and speed congenic backcrossings [17]. The resulting *Cobl* mouse strain was backcrossed to C57BL/6J until at least C57BL/6J::129/SvJ (87.5::12.5) (87.5% + additional contribution from speed congenics reflecting further back-crossings). WT mice were taken alongside KO animals from heterozygous breedings of the *Cobl* KO strain or were directly taken from C57BL/6J breedings.

Mice were kept with ad libitum access to food and water at room temperature (22 °C) and with 68% humidity in a 14 h light/10 h dark cycle in a specific pathogen-free environment, which was regularly verified on sentinel animals housed under identical conditions.

For PCR and Western blot analyses, organs and tissues were removed from mice after killing, quick-frozen in liquid nitrogen and stored at −80 °C.

All animal care and experimental procedures were performed in accordance with the EU animal welfare protection laws and regulations and were approved by a licensing committee from the local government under the permission numbers UKJ-17-021 (Landesamt für Verbraucherschutz, Bad Langensalza, Thuringia, Germany) and twz19-2017 (Stabsstelle Tierschutz, University Hospital, Jena, Thuringia, Germany).

### 2.2. RNA Isolation, cDNA Synthesis and RT-PCR

RNA isolations were performed as previously described [12]. In detail, tissue samples were transferred from −80 °C to liquid nitrogen and then crushed in a nitrogen atmosphere using a mortar and a pestle. For RNA extraction, 1 ml TRizol was added per 100 mg tissue and the samples were incubated at room temperature for 5 min. After centrifugation for 10 min at 12,000× *g*, TRizol was removed and chloroform was added for phase separation (0.2 ml per 100 mg tissue). After brief shaking and 2 min incubation at room temperature, centrifugation was performed for 15 min at 12,000× *g* and 4 °C. Subsequently, the upper phase was mixed with isopropanol (0.5 ml per 100 mg tissue) at −20 °C for 15 min and centrifuged at 12,000× *g* for 10 min at 4 °C. The resulting pellet was washed with 75% (*v*/*v*) ethanol in DEPC-H_2_O, dried and resuspended in DEPC-H_2_O. The quality and quantity of the isolated material were assessed spectrophotometrically with NanoDrop OneC (Thermo Fisher Scientific, Inc., Waltham, MA, USA).

Per sample, 50 μg of RNA were treated with 2.5 μL DNase I (3 U/μL) and 10 μL RDD buffer (QIAGEN^®^ GmbH, Hilden, Germany) in a total volume of 100 μL in DEPC-H_2_O for 10 min. RNA was precipitated by adding 11 μL of 4 M LiCl (in DEPC-H_2_O) and 275 μL of 100% precooled (−20 °C) ethanol. After incubation for 30 min at −80 °C, the samples were centrifuged at 14,000× *g* and 4 °C. The sediment was washed with precooled 70% (*v*/*v*) ethanol, dried and dissolved in 100 μL DEPC-H_2_O. RNA concentration was determined spectrophotometrically and the samples were stored at −80 °C.

The RevertAid H Minus First Strand complementary DNA Synthesis Kit (Thermo Fisher Scientific, Inc.) was used to transcribe RNA into cDNA according to procedures described previously [12]. In detail, 2 μg of the RNA was diluted in 11 μL DEPC-H_2_O and 1 μL 0.5 μg/μL oligo(dT)18 primer solution and incubated for 5 min at 65 °C. Subsequently, 4 μL 5x reaction buffer, 1 μL RiboLock RNase Inhibitor, 2 μL 10 mM dNTP mixture and 1 μL reverse transcriptase were added. Negative controls without reverse transcriptase were processed alongside. The reaction mixtures were incubated at 42 °C for 1 h and at 70 °C for 5 min and thereafter stored at −80 °C.

Gene expression was investigated by RT-PCR using exon-spanning primers (*Cobl* 5′-GCTCCGGAAGACTGCAGAACA-3′ (forward), 5′-GGATGACACCTTCCTGAGGC-3′ (reverse); *Gapdh* 5′-ATTGACCTCAACTACATGGTCTACA-3′ (forward), 5′-CCAGTAGACTCCACGACATACTC-3′ (reverse)). For PCR, 0.5 μL cDNA was mixed with 4 μL 5× Phusion HF buffer, 0.4 μL 10 mM dNTPs, 1 μL 10 μM forward and reverse primers, 12.9 μL Ampuwa water and 0.2 μL Phusion DNA Polymerase (2 U/μL; Thermo Fisher Scientific Inc.). Samples were initially denatured at 98 °C for 1 min, followed by 35 cycles for *Cobl* or 30 cycles for *Gapdh* of 10 sec of denaturation at 98 °C, 20 sec of primer hybridization (*Cobl* 62 °C, *Gapdh* 60 °C) and 15 sec of elongation at 72 °C. Final elongation was carried out at 72 °C for 10 min.

### 2.3. In Situ Hybridization

In situ hybridization probes were generated and purified as described earlier [19]. The nucleotides 365–680 of the coding sequence of the murine *Cobl* gene were amplified using the following primers: 5′-GAGAGCTCCGTTGATTGGGTCCTTGAAT-3′ (forward) and 5′-GAGGTACCCTGTT-GTCCCACGCATACAG-3′ (reverse).

Fresh-frozen tissue sections (16 μm) from P0 WT C57BL/6J mice were prepared, hybridized overnight (at 67 °C) and stringently washed to remove unspecific background, as previously reported [19].

Blocking and antibody staining were performed using 5-Bromo-4-chloro-3-indoxylphosphate (Roche, Basel, Switzerland) and nitro blue tetrazolium (Roche) for the detection of alkaline phosphatase-conjugated antibodies according to procedures described previously [19]. Finally, the sections on the slides were washed, post-fixed in 4% (*w*/*v*) paraformaldehyde in PBS and mounted with Mowiol (Calbiochem/Merck KGaA, Darmstadt, Germany). An AxioObserver.Z1 microscope equipped with an ApoTome and a Plan-Apochromat 20×/0.5 objective (Zeiss, Oberkochen, Germany) was used for imaging. The tile function of the ZEN2012 software was used to generate larger tissue overviews.

### 2.4. Immunoblotting Analysis

Frozen tissue was homogenized in RIPA buffer (50 mM Triton X-100, 150 mM NaCl, 1% (*v*/*v*) IGEPAL, 0.5% (*v*/*v*) deoxycholic acid, 0.1% (*w*/*v*) SDS) supplemented with 1× Complete protease inhibitor without EDTA (Roche) and 200 mM Calpain inhibitor 1 (Sigma-Aldrich Co. LLC/Merck KGaA, Darmstadt, Germany) using a Potter S Homogenizer (Sartorius AG, Göttingen, Germany). Homogenates were centrifuged for 15 min at 10,000× *g* and 4 °C. In total, 50 μg protein per sample was used for immunoblotting analysis.

Primary antibodies for immunoblotting analyses included monoclonal mouse anti-β-actin antibodies (Sigma-Aldrich; RRID:AB_476744) and polyclonal guinea pig anti-Cobl^DBY^ antibodies [12].

Secondary antibodies were purchased from Thermo Fisher Scientific Inc. (Alexa Fluor680 goat anti-mouse IgG (RRID:AB_1965956)) and from LI-COR Bioscience (Lincoln, NE, USA) (IRDye^®^ 800CW donkey anti-guinea pig IgG (RRID_AB1850024)).

Detection was carried out in a fluorescence-based manner to ensure the linearity of the signals, as described previously [11,15]. Fluorescence detections were carried out with a Li-COR Odyssey System (LI-COR Bioscience).

### 2.5. Electron Microscopy

After sacrificing P10 WT and *Cobl* KO mice, the soft tissue of the lower jaw was removed and the remaining bone including the teeth was incubated at 4 °C in Karnovsky’s solution [20] until further processing. After 24 h of water immersion, the tissue was dried and subsequently blocked in Epofix (Struers GmbH, Willich, Germany) for another 24 h. Sections of a thickness of 100 μm were prepared in the sawing microtome by fixing the tissue block with Technovit 7210 VLC (Heraeus Kulzer GmbH & Co. KG, Wehrheim, Germany).

After mounting the sections onto holders, the samples were carbon-coated (thickness, 14 nm) to prevent surface charging utilizing a CCU-010 sputter coater (Safematic GmbH, Zizers, Switzerland). Afterward, the samples were analyzed using a scanning electron microscope LEO-1450 (Zeiss) equipped with an EDX system Quantax 200 with X-Flash 5030 detector (Bruker AXS, Billerica, MA, USA).

For imaging, the back-scattered electron mode was used to visualize the enamel pattern. For EDX investigations, compositional measurements were carried out by continuous scanning of surface areas of 20 μm × 40 μm, resulting in local average values.

Due to structural differences in the outer and inner enamel in the P10 thin sections, both enamel layers were measured separately.

### 2.6. Immunofluorescence Analyses of Tissue Sections and Microscopy

Heads of P0 and P10 animals were incubated for 24 h each in 4% (*w*/*v*) paraformaldehyde solution and 10% (*w*/*v*) and 30% (*w*/*v*) sucrose solution at 4 °C. The fixed mouse heads were embedded in dry ice using Tissue-Tek^®^ O.C.T. compound (Sakura Finetek Europe B.V., Alphen aan den Rijn, Netherlands). To avoid splintering of the P10 heads due to the onset of mineralization, P10 tissue was demineralized in 10% (*w*/*v*) EDTA at room temperature for 1 week prior to embedding. Coronal cryosections with a thickness of 12 μm were prepared on a Leica CM3050 S (Leica Microsystems GmbH, Wetzlar, Germany) at −28 °C.

Immunostaining was performed on cryosections of the first to third molars. Sections were washed with 100 mM sodium phosphate buffer pH 7.4 and blocked for 1 h in blocking solution (0.25% (*v*/*v*) Triton X-100, 5% (*v*/*v*) goat serum in 100 mM sodium phosphate buffer pH 7.4). After primary antibody incubation (24 h; 4 °C; in blocking solution), slices were washed and incubated with secondary antibodies, phalloidin and DAPI in blocking solution at room temperature for 2 h in the dark, washed again and mounted in Fluoromount G (SouthernBiotech, Birmingham, AL, USA).

Primary antibodies included polyclonal guinea pig anti-Cobl^DBY^ antibodies [12] and polyclonal rabbit anti-amelogenin antibodies (Millipore/Merck KGaA, Darmstadt, Germany; ABT260). Secondary antibodies were purchased from Thermo Fisher Scientific Inc. (Alexa Fluor^®^ 488 goat anti-guinea pig IgG, RRID:AB_2534117; Alexa Fluor^®^ 568 donkey anti-rabbit IgG, RRID: RRID:AB_2783823). Phalloidin AlexaFluor^TM^ 647 was used to visualize F-actin (Thermo Fisher Scientific Inc.) and DAPI to highlight nuclei (Roche).

Immunofluorescence signals were imaged using a Zeiss Axio Observer Z1 microscope/Apotome (objective, Plan-Apochromat 63×/1.4) and an AxioCam MRm CCD camera (Zeiss). Digital images were recorded by ZEN2012 (RRID SCR_013672). Image processing was carried out by ImageJ (RRID:SCR_003070) or Adobe Photoshop (Adobe, San Jose, CA, USA) (RRID: SCR_014199). Image processing was performed equally and did not include any gamma adjustments.

### 2.7. Examinations of Ameloblast Height and of Ameloblast Actin Cytoskeleton

The ameloblast height was determined on phalloidin-stained tooth germs of mandi-bular molars by measuring the distance between the apical and basal membranes of the enamel-forming cells using ImageJ. The crown of each coronally incised tooth was divided into 6 segments (fissure–cusp, cusp and cusp–equator, twice each), with up to 5 measurements being carried out in each region. To additionally clarify the change from P0 to P10, the ameloblast heights of P10 were determined as a percentage of P0.

F-actin intensities were determined by phalloidin staining and confocal microscopy. Fluorescence intensity measurements were conducted at the apical membrane area of the ameloblasts of mandibular molars. Measurements were carried out over a 30 μm distance from the enamel matrix to the cytosol of the ameloblasts. The center of these measurements was placed at the apical membrane itself. Up to 5 randomly placed measurements were taken within each of the 6 regions of the dental crown.

The apical membrane was additionally assessed separately. Phalloidin fluorescence measurements were conducted from positions ranging from −1 μm to +1 μm and averaged.

### 2.8. Measurement of the Enamel Area at P10

Images of mandibular molars were taken in transmitted light mode using the Zeiss Axio Observer Z1 microscope, AxioCam MRm CCD camera and ZEN2012 (Zeiss). The enamel area and the area of the tooth crown were measured using ImageJ by outlining the enamel area as well as the apical enamel extensions.

### 2.9. Experimental Design and Statistical Analyses

No statistical methods were used to predetermine the sample size. All quantitative evaluation data were taken into account to completely represent biological and technical variabilities.

Tests for normal data distribution and statistical significance calculations were performed using Prism 10 software (GraphPad, La Jolla, CA, USA) (RRID:SCR_002798). Statistical tests applied are specified in the figure legends. Statistical significances are marked by * *p* < 0.05, ** *p* < 0.01, *** *p* < 0.001 and **** *p* < 0.0001 throughout. Whenever possible, exact *p* values are reported in the figures.

## 3. Results

### 3.1. The Actin Nucleator Cobl Is Expressed in Murine Jaws

In situ hybridizations of sections of heads of P0 mice detected pronounced levels of mRNA of the actin nucleator Cobl in the retina, the olfactory bulbs and in the nasal cavity (Figure 1A). All of these localizations may reflect known Cobl expressions in the nervous system and in sensory organs [9,17,21]. Strikingly, *Cobl* mRNA was also very abundant in mandibular molars (Figure 1A). Also, in maxillary molars, *Cobl* expression was detected (Figure 1A).

Interestingly, the expression of *Cobl* mRNA was most readily detectable in the developing tooth crown areas (Figure 1A). Analyses at higher magnifications demonstrated that the *Cobl* mRNA occurrence in developing teeth clearly represented a high expression of mRNA of the actin nucleator *Cobl* in the ameloblast cell layer (Figure 1A; inset).

Reverse transcriptase polymerase chain reactions (RT-PCRs) showed that *Cobl* mRNA could not only be detected in the brain [9,12,17] and brain-containing samples, such as full heads that served as positive controls, but also in mRNA isolated from head samples without the brain (Figure 1B,C). Importantly, *Cobl* mRNA expression was detected in upper jaw (maxillary) and lower jaw (mandibular) samples from mice at ages ranging from E14 to adulthood (Figure 1B,C).

Examinations at the mRNA level thus demonstrate *Cobl* mRNA expression in the tooth-bearing jaws over the period of amelogenesis.

Cobl protein expression analyses at the developmental time point P0 representing the secretory stage of amelogenesis specifically detected Cobl not only in brain-containing tissue samples but also in heads without brains and, importantly, also specifically in maxilla and mandible samples (Figure 1D). In order to follow the protein expression of Cobl in the tooth-bearing jaws over the period of enamel development, upper (Figure 1E) and lower jaws (Figure 1F) were prepared from E14, P0, P5, P10 and P18 mice. Cobl was expressed in both upper and lower jaw over the whole period of amelogenesis (Figure 1E,F). The expression tended to decrease over the course of enamel development (Figure 1E,F).

Our observations of Cobl expression in ameloblasts were in line with two transcriptome analyses of murine enamel organs during the secretion and maturation phase of ameloblasts, in which *Cobl* mRNA was detected [22,23]. Also, these transcriptome studies suggested that *Cobl* expression may be higher during the secretion phase [22,23].

### 3.2. The Cobl Protein Can Be Detected in the Tooth Germ of a Murine Mandibular Molar at Different Stages of Amelogenesis

In order to obtain additional support for the hypothesis that the actin nucleator Cobl may play a thus far unknown role during amelogenesis, we next examined the expression of Cobl in the tooth germ by immunofluorescence microscopy of coronally sectioned mouse heads in the secretory stage (P0) and maturation stage (P10) (Figure 2).

At P0 the mandibular tooth germ was localized in the alveolar process in the immediate vicinity of the oral cavity and tongue. The nuclei of the ameloblast layer, which extended over the tooth crown, were always located basally. The enamel-forming ameloblasts were arranged in a palisade pattern (Figure 2A). Anti-amelogenin immunostaining highlighted the function of the tooth germ in the secretion phase with the secretion of the matrix proteins, which began in the cusp area. Closer examination of the cusp region revealed anti-amelogenin signals inside the ameloblasts and a thin layer of anti-amelogenin immunoreactivity that represents already secreted enamel matrix material (Figure 2B). F-actin was present at the cell cortex of the ameloblasts and was particularly localized apically and basally (Figure 2C). The actin nucleator Cobl was detected basally, apically and in the apical cytosol (Figure 2D). Basal and apical enrichments were thus consistently observed for both F-actin and Cobl in P0 ameloblasts (Figure 2C–E). The Cobl signal could also be detected in the surrounding tissue, for example, in the bone of the alveolar process (Figure 2D,E). These observations were in line with the in situ hybridizations also showing some expression of *Cobl* in the tissue surrounding the tooth germ (Figure 1).

At P10, representing the maturation phase, the tooth germ had increased in size (Figure 2F). The ameloblasts were much shorter and now had a cubic shape. In contrast to P0, anti-amelogenin immunosignals were only partially detectable in the maturing enamel. This reflected the demineralization process, which caused a partial dissolution of the maturing enamel (Figure 2G). Similar to P0, Cobl was localized in a pattern very similar to F-actin. Both were enriched basally, in the apical cytosol and in particular close to the apical plasma membrane of the ameloblasts at P10 (Figure 2H–J).

The close proximity of Cobl to F-actin filaments in ameloblasts during enamel formation was in line with some actin cytoskeletal role of Cobl in amelogenesis.

### 3.3. Developing Mandibular Molars of Cobl KO Mice Show an Increased Carbon Content in the Enamel When Compared to WT Molars

In order to address whether a lack of the actin nucleator Cobl, which showed high protein expression in the ameloblasts during tooth germ development, manifests in an altered elemental composition of the enamel during amelogenesis, we next examined the enamel of developing mandibulars with energy dispersive X-ray spectroscopy (EDX) [24]. Scanning electron microscopic analyses showed optical structural differences in the inner and outer enamel of P10 mandibular molars that may reflect different stages of enamel maturation or enamel organization (Figure 3A). In order to account for putative differences between these two zones, we conducted separate EDX examinations of the outer and inner enamel. Characteristic X-rays permitted the detection of the elements calcium (Ca), phosphorus (P), oxygen (O), carbon (C) and sulphur (S) in the enamel of the molars (Figure 3B).

Quantitative analyses comparing WT and *Cobl* KO mice uncovered that the mean carbon content of the entire enamel under Cobl deficiency was significantly increased by 20% when compared to WT (Figure 3C; ** *p* = 0.0027). This increase in carbon content in *Cobl* KO enamel seemed to come at the expense of calcium, oxygen and phosphate representing the main elements of hydroxylapatite (Figure 3C).

The phenotype of the increased carbon content in the enamel of *Cobl* KO mice was also evident when analyzing the enamel layers separately, as inner and outer enamel areas showed similar element compositions (Figure 3D,E). Despite the lower n numbers in the separate averagings, the increased carbon content reached statistical significance in the inner enamel of *Cobl* KO compared to WT enamel (Figure 3E).

### 3.4. The Absence of Cobl Leads to an Increased Enamel Surface of Developing Mandibular Molars

In order to investigate whether the altered composition of developing mandibular molars under Cobl deficiency would be accompanied by alterations in the enamel quantity, the area of the maturing enamel of mandibular molars was quantitatively measured at the developmental stage P10 using unstained cryosections (Figure 3F,G). The analysis showed that the ratio of enamel area to tooth crown area was significantly increased in the *Cobl* KO compared to the WT (Figure 3H). These measurements demonstrated that not only the elementary composition but also the enamel area was altered in P10 *Cobl* KO mice when compared to WT mice. The observed increase in the enamel and the increased carbon content suggest that *Cobl* KO causes defects in ameloblast-mediated enamel maturation.

### 3.5. The Height of Ameloblasts During Amelogenesis Is Increased upon Cobl Deficiency

The identified changes in the enamel of *Cobl* KO mice raised the question, which cellular defects are linked to the observed ameloblast dysfunction in proper enamel formation. We therefore analyzed whether cellular aspects of amelogenesis may be impaired by *Cobl* KO during either the secretory phase (P0) and/or the maturation stage (P10).

To test this, the distance between the apical and basal membranes of the ameloblasts was determined in phalloidin-stained cryosections of developing mandibular molars. In these examinations, the tooth crown was divided into the three areas: fissure–cusp, cusp and cusp–equator (Figure 4A–E).

In the secretory stage (P0), the ameloblast height was highly significantly increased upon *Cobl* KO when compared to WT. With 24.3%, the percentage increase in the height of *Cobl* KO ameloblasts compared to WT ameloblasts was most pronounced in the cusp–equator area (Figure 4F; ****, *p* < 0.0001). Also, in the cusp area (+13.5%; Figure 4G; ***, *p* = 0.0004) and in the fissure–cusp area (+12.7%; Figure 4H; ***, *p* = 0.0001), the increases in ameloblast heights were highly statistically significant. During the maturation stage (P10), the deviations between *Cobl* KO and WT were less pronounced. In the cusp area, the ameloblast height was significantly increased by 8.8% upon Cobl deficiency compared to WT (Figure 4I–K). The cell morphology of the ameloblasts is thus influenced by Cobl deficiency.

To explicitly address the process of shortening of the enamel-forming cells from P0 to P10, the ameloblast heights of P10 were determined as a percentage of P0. As already obvious by comparing the strong phenotypes in length control in P0 ameloblasts versus the almost complete lack of differences at P10, the process of ameloblast shortening was not negatively affected in *Cobl* KO mice—on the contrary, the percental extent of shrinkage was even more pronounced in *Cobl* KO ameloblasts in the cusp–equator and in the fissure–cusp area (Figure 4L-N). In the cusp–equator region, this effect was particularly strong (−21.1% compared to WT) and highly statistically significant (****, *p* < 0.0001) (Figure 4L).

The cell morphological adaptations underlying the functions of ameloblasts during amelogenesis are thus significantly impaired by Cobl deficiency.

### 3.6. Developing Ameloblasts of Cobl KO Mice Show an Altered Mean F-Actin Intensity in the Area of the Apical Membrane When Compared to WT Mice

The impaired ameloblast morphology and the defects in enamel maturation observed upon Cobl deficiency may relate to putative changes in actin cytoskeletal structures underlying the apical plasma membrane region, across which numerous transport processes have to occur during enamel secretion and maturation [25]. To follow up on this hypothesis, the F-actin intensity was determined on the apical membrane side of mandibular molar ameloblasts at the secretory (P0) and maturation stage (P10) based on phalloidin staining.

At the cusp–equator region at P0, a spatially resolved measurement of the mean F-actin intensity over the course of a 30 μm long measurement path perpendicular to the apical membrane showed no gross differences in mean F-actin intensity (Figure 5A). However, a separate analysis of the area directly at the apical membrane showed a small but statistically significant increase in the mean F-actin intensity in this apical zone under Cobl deficiency compared to the WT (Figure 5B).

In the cusp region, the mean F-actin intensity in *Cobl* KO was highly significantly increased over the course of the apical membrane region (−1.02 μm to +1.33 μm) (Figure 5C). This observation was likewise evident when the narrow portion directly at the apical membrane region was determined and averaged separately. With plus 16.4% in *Cobl* KO compared to WT, the percentage difference was greatest in the cusp region (Figure 5D). In the fissure–cusp region, the mean F-actin intensity in the apical, subcortical cytosol (+1.84 μm to +2.99 μm) of the ameloblasts was significantly lower in *Cobl* KO compared to WT (Figure 5E,F).

The investigation of the developmental time point P10, the maturation stage of ameloblasts, showed a completely different phenotype at the apical membrane side of the ameloblasts (Figure 6A–F). During this time of ameloblast and enamel maturation, gross actin cytoskeletal defects were observed. In particular, the actin cytoskeleton in the apical membrane zone of the ameloblasts was strongly impaired upon *Cobl* KO and showed a dramatic lack of F-actin (Figure 6A–F).

Quantitative examinations revealed a highly significant reduction in mean F-actin intensity in the *Cobl* KO in the apical membrane area of all three zones of the molar crown, i.e., in the cusp–equator (−6.33 μm to +3.57 μm; Figure 6A), in the cusp (−2.65 μm to +3.06 μm; Figure 6C) and in the fissure–cusp (−2.86 μm to +2.14 μm; Figure 6E) regions when compared to the WT. Detailed evaluations of specifically the apical membrane region (−1 μm to +1 μm) demonstrated that the percentage loss in *Cobl* KO was greatest in the cusp–equator region with minus 33.4% (Figure 6B), followed by minus 28.7% in the cusp region (Figure 6D) and minus 25.1% in the fissure–cusp region compared to the WT (Figure 6F). Only in the cytosol of the cusp–equator region (+4.49 μm to +15 μm), the intensity of F-actin was increased under Cobl deficiency compared to the WT (Figure 6A) but this effect did not occur in the cusp and fissure–cusp regions and also was much more modest when compared to the gross impairments identified at the apical membrane in all three subregions of the developing molars (Figure 6A–F).

*Cobl* KO thus does not only have an influence on cell morphology but also changes the F-actin content in the apical membrane zone of ameloblasts in dependence on their developmental stage and in particular massively impairs the F-actin organization in the ameloblast maturation stage.

## 4. Discussion

The present study provides the first insights into the function of the actin nucleator Cobl in enamel formation and uncovers that Cobl deficiency during amelogenesis in mice affects not only the amount and composition of enamel formed but also the ameloblast morphology and the actin cytoskeleton of ameloblasts.

Our data are in line with a general importance of F-actin during amelogenesis. The cell morphology of the enamel-forming cells and their movement in the ameloblast layer are dependent on actin filaments [4]. The role of Cobl in the process of enamel formation was previously unknown. Our comprehensive analyses showed *Cobl* mRNA expression in ameloblasts in mandibular molars and incisors by in situ hybridizations in P0 mice. In line with a Cobl expression in these specialized cells, transcriptome analyses performed on murine enamel organs also detected *Cobl* mRNA and suggested an increased *Cobl* mRNA expression during enamel secretion compared to the maturation stage [22,23]. Analyzing a broader range of developmental stages, we detected *Cobl* mRNA in the tooth-bearing jaws within the period examined during amelogenesis (E14 to P18) and beyond for the adult stage.

In addition to this work at the mRNA level, our work clearly demonstrates the translation of *Cobl* mRNA transcripts into Cobl protein in murine mandibles. Cobl protein was detected over the entire period of amelogenesis, whereby the Cobl protein was more strongly expressed during the secretory stage. Immunofluorescence analyses of tissue sections showed that Cobl was localized in the region of the apical and basal membrane as well as in the apical cytosol in the ameloblasts of mandibular molars during the secretory and maturation stages with comparably higher expression at the secretory stage. This similar localization pattern compared to F-actin in the ameloblasts suggested that the actin nucleator Cobl may play a role in the regulation of the actin cytoskeleton in ameloblasts of mandibular tooth germs.

Amelogenesis is a complex process. Even the smallest deviations can result in *Amelogenesis imperfecta* with hypoplastic and/or hypomineralized enamel depriving the tooth of its protective barrier against mechanical, thermal and chemical stress. In the long term, this may lead to considerable damage to the dentin and pulp and ultimately to tooth loss [26,27,28]. To investigate whether Cobl deficiency affects murine enamel, enamel composition and enamel quantity were determined. Analysis of the elemental composition of maturing enamel using EDX demonstrated that the enamel of Cobl-deficient mice had a significantly increased carbon content at P10 compared to the WT. An altered composition of the enamel can cause hypomineralization and be associated with enamel defects [26,27]. Mouse jaws at P14 exhibited a significantly reduced enamel volume in MMP20- and KLK4-deficient mice compared to WT. In addition, EDX studies on seven-week-old mandibular incisors showed reduced calcium and increased phosphorus content in the outer enamel [26]. Núñez et al. [27] reported enamel defects in seven-week-old amelotin-deficient mice due to reduced enamel mineralization compared to WT mice. During enamel maturation, enamel proteins are removed from the matrix and ions are secreted, which leads to a decrease in organic and an increase in inorganic components [1,3,25]. The increased carbon content in *Cobl* KO therefore indicates an increased proportion of enamel proteins and consequently a delayed enamel maturation.

The development of hypoplastic enamel has been described for some genetic defects. Deficiency of amelogenin, for example, resulted in a 10–20% reduction in enamel thickness [1], which may be due to reduced enamel formation during amelogenesis. Examinations of the enamel areas of mandibular molars of WT and *Cobl* KO mice at the age of P10 showed that the relative enamel area during amelogenesis was increased under Cobl deficiency compared to WT. This could theoretically reflect some expansion of enamel volume due to defects in maturation and/or represent an increase in the amount of secreted enamel in the *Cobl* KO tooth germs.

During enamel development, a functional actin cytoskeleton controls the cell morphology of the ameloblasts and their movement in the ameloblast layer, which produces the characteristic intertwining of the enamel prisms [6]. Earlier studies on the developing tooth germ have uncovered some correlations between an incorrectly regulated actin cytoskeleton and impaired enamel formation. Enamel defects in the cusp region and reduced enamel thickness were observed in transgenic mice with dominant-negative RhoA [29]. The epithelial-specific deletion of the GTPase Cdc42 was also shown to impair enamel formation in murine molars. In these mice, a reduction in desmosomes and deviating actin filament arrangements was accompanied by a formation of cystic lesions in the enamel organ of the tooth germ [30].

The present study uncovered the influence of Cobl on the ameloblast height and the F-actin intensity in the apical membrane area during enamel secretion and maturation. In the secretion stage (P0), *Cobl* KO showed a generally increased ameloblast height of the mandibular molars in all examined regions of the tooth crown compared to the WT. In addition, the F-actin intensity in the region of the apical membrane was modestly increased, particularly in the cusp area. During the maturation stage (P10), the ameloblast height of the mandibular molars continued to show an increase in the cusp region in the *Cobl* KO. Importantly, the analysis of the apical membrane area showed a significantly reduced F-actin intensity in the *Cobl* KO compared to the WT in all examined regions of the tooth crown. These results suggest that the actin nucleator has different developmental effects on ameloblasts during amelogenesis and reveal massive defects in the actin cytoskeletal organization at the apical membrane of ameloblasts during the maturation phase.

The generally increased ameloblast height and the regionally increased F-actin intensity in the apical membrane area observed in *Cobl* KO mice at the developmental time point P0 may appear counterintuitive considering that the actin nucleator Cobl promotes the formation of actin filaments already at low concentrations [9] and is a positive regulator of dendritogenesis and regrowth of the dendritic tree after stroke [9,13]. However, interestingly, similar effects on cellular morphology were observed in the duodenum, where *Cobl* KO caused an increase in the length of the microvilli [16]. Such phenotypes may reflect the fact that the actin nucleator Cobl could also inhibit filament formation processes essential for ameloblast height and apical membrane organization by sequestering polymerizable G-actin through simple binding of its three WH2 domains to G-actin [9,31,32] if Cobl is not activated for additional nucleation. The absence of Cobl would then make more actin monomers available, which could be used for Cobl-independent actin nucleation and polymerization processes in the secretory stage. Since Cobl and certain fragments thereof have been described to have additional actin-filament-severing activities in in vitro assays [31,32,33], Cobl may have multifunctional, in part even contrary functionalities towards actin filament dynamics.

For actin nucleators, cooperation with each other or compensation in the event of failure may also be relevant mechanisms in the regulation of the actin cytoskeleton. In the context of dendritogenesis, it has already been described that the functionality of Cobl depends on the Arp2/3 complex [11] and furthermore involves Cobl-like—a related component that also promotes F-actin formation [15,19]. Related cooperations and/or compensations may also be conceivable for ameloblasts in order to ensure a regular course of amelogenesis.

Among the different actin nucleators identified, Cobl stands out as an evolutionary rather young nucleator with rather specialized functions in a distinct set of specialized cells. Whereas the ur form of the Cobl protein family, Cobl-like, already evolved with the emergence of bilateria, a vertebrate-specific gene duplication evolved the *Cobl* gene, gaining two additional WH2 domains leading to the actin nucleating Cobl [34]. High-resolution structural analyses [35] have furthermore suggested unique features of the Cobl-mediated actin nucleation mechanism in comparison to other WH2-domain-based actin nucleators such as JMY [36] and spire [37]. Importantly, Cobl thereby seems to be particularly responsible for rather unique, specialized cytoskeletal structures, such as fine structures in the terminal web of enterocytes connecting the terminal web underlying the plasma membrane with microvilli rootlets [16]. Also, in early postnatal outer hair cells in the cochlea, Cobl was shown to be important for a specialized set of actin filaments in the cuticular plate supporting the stereociliar bundle [17]. In a different mouse model, *Cobl*^Δ84/Δ84^ mice showed reduced junctional F-actin accumulation within the circumferential ring in epithelial cells and impaired epithelial paracellular barrier function in the stomach [33]. Cobl was also reported to be required for the maintenance of the apical actin network in mature, but not immature ependymal cells [38]. These observations in other specialized cells seem to be mechanistically somewhat related to the strongly impaired apical actin cytoskeleton in P10 ameloblasts of *Cobl* KO mice observed during our examinations of tooth formation.

The developmental differences in phenotypes observed suggest that Cobl may take up its cell biological function as an actin nucleator with some delay. Accordingly, at P0, Cobl’s inhibitory effect on actin filament formation by quenching G-actin may outweigh positive contributions to F-actin formation and manifests in a reduced ameloblast height in all regions as well as in a regionally reduced F-actin intensity in the apical membrane area. At the maturation stage (P10), Cobl’s G-actin quenching effect seems to have a lower impact, but instead, Cobl’s F-actin-promoting role by powering actin nucleation seems to be critically required, as reflected in the massively decreased F-actin intensity in the apical membrane area in ameloblasts of *Cobl* KO mice.

Possibly, the influence of binding partners and/or regulators of Cobl could change the functionality of Cobl from the secretory stage (P0) to the maturation stage (P10). For example, the interaction with syndapins seems to be modulating the recruitment of the actin nucleator Cobl to the plasma membrane, as shown in the context of neuromorphogenesis [11]. The evolutionary ancestor of Cobl, Cobl-like [19], which had been detected at the mRNA level in the enamel organ [23], may compete and/or cooperate with Cobl as in neuromorphogenesis [15]. Regulation of Cobl by PRMT2 could control the functionality of the actin nucleator via arginine methylation of the C-terminal actin nucleation domain, as demonstrated in neuronal cells [18]. Likewise, a temporally distinct control of Cobl via calcium-activated calmodulin [14] may underlie the phenotype transformation in ameloblasts.

Similar considerations may also explain the effects of Cobl deficiency on the tooth enamel in mandibular molars. Since an increased enamel thickness was observed during amelogenesis in *Cobl* KO mice when compared to WT mice, Cobl functions seem to exert an inhibitory influence on the processes of enamel secretion by reducing the release of enamel matrix proteins. Consistently, the investigation of the elemental enamel composition of mandibular molars at P10 revealed a delayed enamel maturation, as the carbon content in the *Cobl* KO enamel was increased when compared to the WT. Cobl could have a promoting influence on the processes of enamel maturation so that the degradation and removal of matrix proteins and the secretion of enamel minerals proceed more quickly. The greater shortening of the enamel-forming cells from P0 to P10 under Cobl deficiency could also represent changes in the functions of ameloblasts and have a lasting effect on enamel formation.

Our study thus revealed that the actin nucleator Cobl is required for proper expansion and mineralization of tooth enamel and these findings are related to development-dependent Cobl functions in ameloblast morphology control and in apical F-actin organization in ameloblasts. The observed local and temporal differences in the *Cobl* KO phenotypes hereby are related to the fact that also the processes of enamel secretion and maturation are complex and do not occur simultaneously within the developing tooth crown.

## Figures and Tables

**Figure 1 cells-14-00359-f001:**
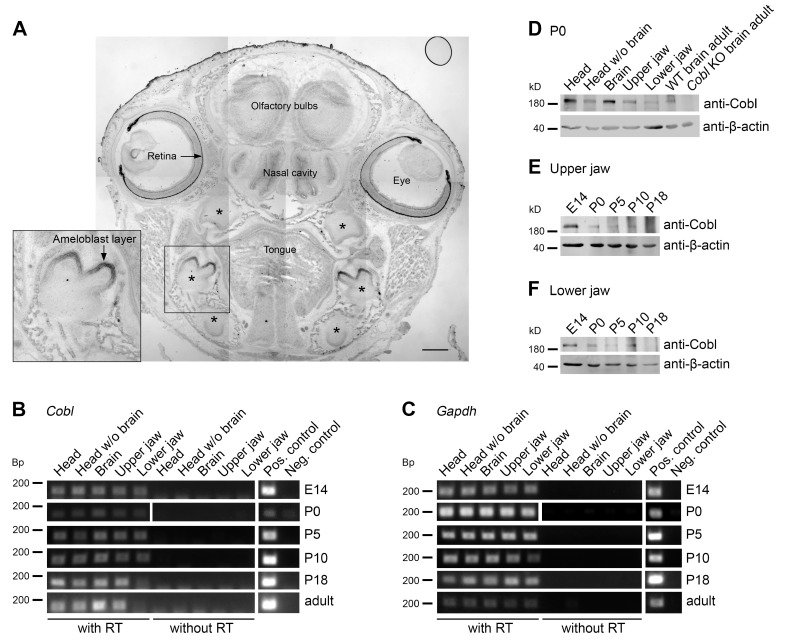
Cobl is expressed in murine jaw during enamel development. (**A**) Tiled overview image of an in situ hybridization of a coronal section of a WT P0 mouse head with a *Cobl* antisense RNA probe. Note the detection of *Cobl* mRNA in the retina, the olfactory bulbs and in the nasal cavity as well as in the mandibular molars, the maxillary molars and the incisors (asterisks). The *Cobl* expression in the mandibular molars is additionally presented as magnified inset. The *Cobl* mRNA signal is likely to represent the ameloblast cell layer of particularly the developing tooth crown. Bar, 500 μm. (**B**,**C**) RT-PCR analyses show the expression of *Cobl* (**B**) in comparison to *Gapdh* (**C**) in maxillary and mandibular samples during the period of amelogenesis, exemplarily for the developmental time points E14, P0, P5, P10, P18 and adult (47 weeks). In addition to the jaw samples, the tissues head, head without brain and brain were examined. *Cobl* and *Gapdh* were specifically detectable in the samples with reverse transcriptase (RT) at all age points. (**D**) Immunoblots of different tissue homogenates at P0 (with full head and brain of WT mice serving as positive controls and brain from *Cobl* KO mice serving as negative control). (**E**,**F**) Developmental series (E14-P18) of anti-Cobl immunoblottings of the maxilla (upper jaw; (**E**)) and mandible (lower jaw; (**F**)). Note that at P0, the actin nucleator Cobl was detectable in all WT tissue samples (**D**), and that Cobl expression tends to be highest at E14 and P0 and to decrease over time (**E**,**F**). β-actin immunoblottings are shown for comparison.

**Figure 2 cells-14-00359-f002:**
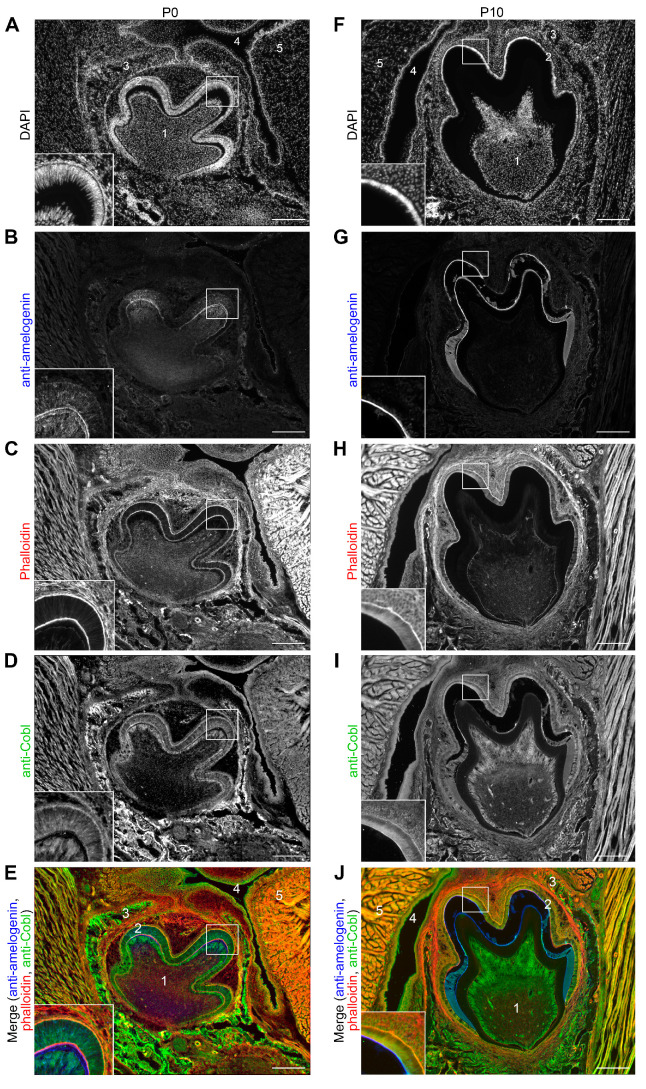
Localization of the Cobl protein in the tooth germ of murine mandibular molars in the secretory (P0) and maturation stage (P10) of amelogenesis. Exemplary, individual z plane Apotome images of coronally sectioned tooth germs of mandibular molars of WT mice at P0 (**A**–**E**) and at P10 (**F**–**J**), respectively. Insets represent enlargements of the boxed areas (parts of cusp area with ameloblast layer). In (**A**,**E**,**F**,**J**), the mandibular tooth germ (1) with the surrounding ameloblast layer (2), which was localized in the alveolar process (3) in anatomical proximity to the oral cavity (4), and the tongue (5), are marked. DAPI staining (**A**,**F**) revealed the DNA of the basally located nuclei of the ameloblasts, which appeared in a palisade-shaped pattern at P0 and in a more cubic shape at P10. Anti-amelogenin immunostaining (**B**,**G**) showed an enrichment of amelogenin inside of P0 ameloblasts (especially in the cusp region) and in the enamel matrix (**B**). At P10, anti-amelogenin immunostaining showed the remaining matrix protein after demineralization (**G**). Phalloidin staining detected F-actin particularly apically and basally in the ameloblasts (**C**,**H**). (**D**,**E**,**I**,**J**) Anti-Cobl immunostainings (**D**,**I**) and merges of anti-Cobl (green), phalloidin (red) and anti-amelogenin (blue) fluorescence signals (**E**,**J**) showing overlapping localizations of F-actin with Cobl, which was enriched basally, in the apical cytosol and in the region of the apical membrane of ameloblasts. In addition, the Cobl signal was detectable in the surrounding tissue and, also in these places, mostly coincided with the phalloidin staining (**C**–**E**,**H**–**J**). Bars, 200 μm.

**Figure 3 cells-14-00359-f003:**
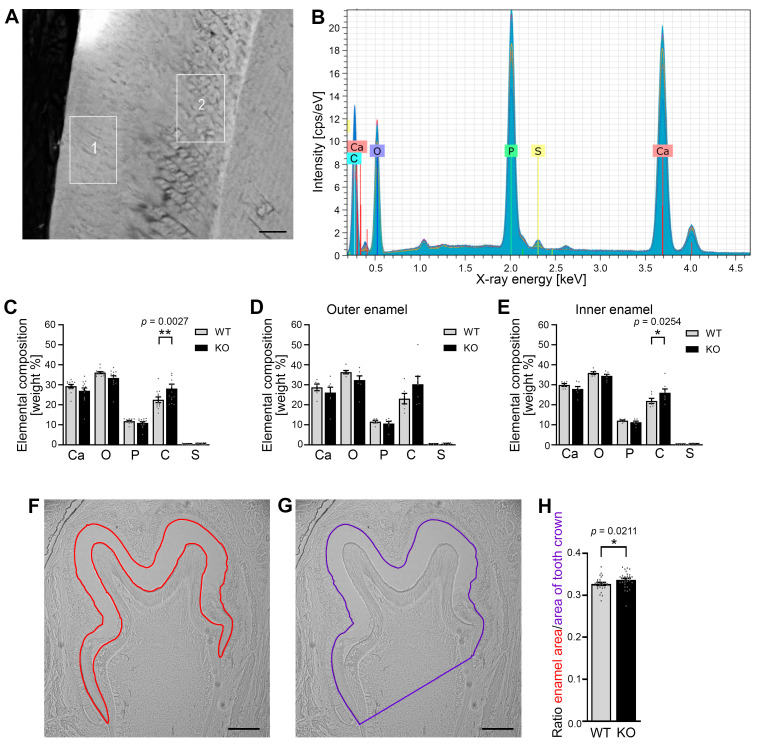
The carbon content and the ratio of enamel to tooth crown area in the enamel of murine mandibular molars at P10 were increased in *Cobl* KO compared to the WT. (**A**) Scanning electron microscope image of the enamel of murine mandibular molars at P10. An area of 20 × 40 μm was measured in both the outer (1) and the inner enamel (2). Bar, 10 μm. (**B**–**E**) EDX spectrum of the enamel of a mandibular molar at time P10 (**B**) and the comparison of the elemental composition between Cobl-deficient and WT mice (**C**–**E**). The X-ray spectrum shows characteristic peaks of calcium (Ca), oxygen (O), phosphorus (P), carbon (C) and sulphur (S) (**B**). The mean values of the proportions of the elements were quantified for the entire enamel (**C**) as well as for outer (**D**) and inner enamel (**E**) in weight percent for WT and *Cobl* KO. The carbon content was increased under Cobl deficiency compared to the WT, with significant differences in the total (**C**) and inner enamel (**E**). (**F**,**G**) Exemplary light microscopy images showing the determination of the enamel area ((**F**), red) and the area of the tooth crown ((**G**), purple) of a first mandibular molar of a WT mouse at P10. Bars, 150 μm. (**H**) Quantitative analysis of the ratio of the enamel area to the tooth crown area of mandibular molars for WT and *Cobl* KO mice at the age of P10. The ratio in developing *Cobl* KO mandibular molars was significantly higher than in the WT. (**C**) WT, n = 12, *Cobl* KO, n = 14 measurements of enamel areas (from molars of 2 animals each). (**D**,**E**) WT, n = 6, *Cobl* KO, n = 7 measurements. (**H**) WT, n = 26, *Cobl* KO, n = 30 mandibular molars (from 3 animals for each condition). Data mean ± SEM. Bar/dot plots. A two-way ANOVA followed by a Šídák’s multiple comparisons test (**C**–**E**) and a Mann–Whitney test (**H**) was performed for statistical analysis. *, *p* < 0.05; **, *p* < 0.01.

**Figure 4 cells-14-00359-f004:**
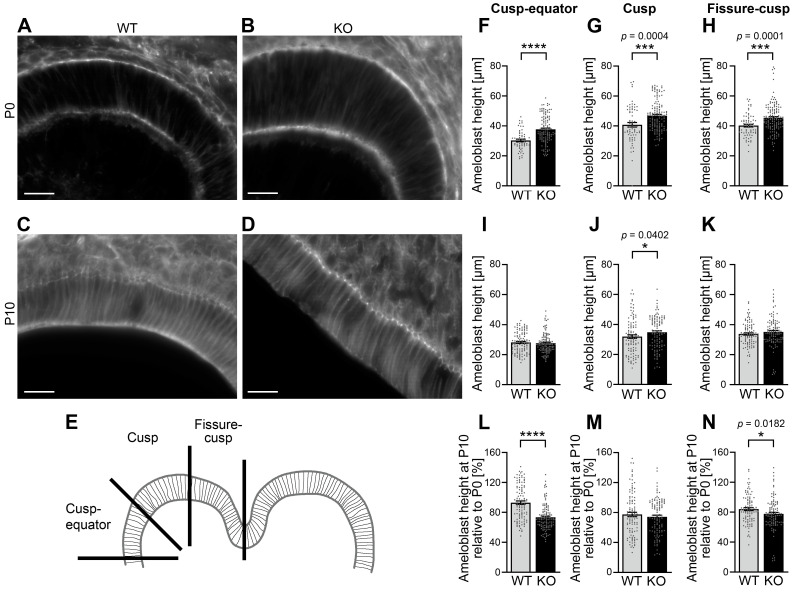
Cobl deficiency causes an increase in ameloblast height during amelogenesis. (**A**–**D**) Exemplary, individual z plane Apotome images showing phalloidin-stained ameloblasts of mandibular molars of WT (**A**,**C**) and *Cobl* KO mice (**B**,**D**) at P0 (**A**,**B**) and P10 (**C**,**D**), respectively. Bars, 20 μm. (**E**) Depicted is a coronal tooth germ of a mandibular molar. Each cusp was divided into three areas: fissure–cusp, cusp and cusp–equator. (**F**–**K**). The comparison of the ameloblast height of murine mandibular molars between *Cobl* KO and WT at the time points P0 (**F**–**H**) and P10 (**I**–**K**) is shown, whereby the distance between the apical and basal membrane was measured based on phalloidin-stained cryosections. This was subdivided into the areas cusp–equator (**F**,**I**), cusp (**G**,**J**), and fissure–cusp (**H**,**K**). At P0 the ameloblast height in *Cobl* KO was highly significantly increased compared to WT (**F**–**H**). At P10, the differences between the genotypes were less pronounced (**I**–**K**). Ameloblast height was still significantly increased in the cusp region under Cobl deficiency (**J**). (**L**–**N**) The ameloblast height of murine mandibular molars at time P10 as a percentage of P0 for *Cobl* KO and WT in the cusp–equator (**L**), cusp (**M**) and fissure–cusp (**N**) regions shows that the percentage of ameloblast height of P10 relative to P0 was significantly reduced in the cusp–equator and fissure–cusp regions in *Cobl* KO compared to the WT. (**F**), WT, n = 56 measurements (12 sections; 7 molars; 3 animals), *Cobl* KO, n = 122 (25 sections; 15 molars; 5 animals). (**G**), WT, n = 69 measurements (14 sections; 7 molars; 3 animals); *Cobl* KO, n = 140 (28 sections; 15 molars; 5 animals). (**H**), WT, n = 69 measurements (14 sections; 7 molars; 3 animals); *Cobl* KO, n = 140 (28 sections; 15 molars; 5 animals). (**I**), WT, n = 110 measurements (22 sections; 12 molars; 3 animals); *Cobl* KO, n = 120 measurements (24 sections; 12 molars; 5 animals). (**J**), WT, n = 102 measurements (22 sections; 12 molars; 3 animals); *Cobl* KO, n = 114 measurements (23 sections; 12 molars; 5 animals). (**K**), WT, n = 100 measurements (20 sections; 12 molars; 3 animals); *Cobl* KO, n = 108 measurements (23 sections; 12 molars; 5 animals)). (**L**–**N**), n numbers as in (**F**–**K)**. Data mean ± SEM. Bar/dot plots. Statistical evaluations, Mann–Whitney tests. *, *p* < 0.05; ***, *p* < 0.001; ****, *p* < 0.0001.

**Figure 5 cells-14-00359-f005:**
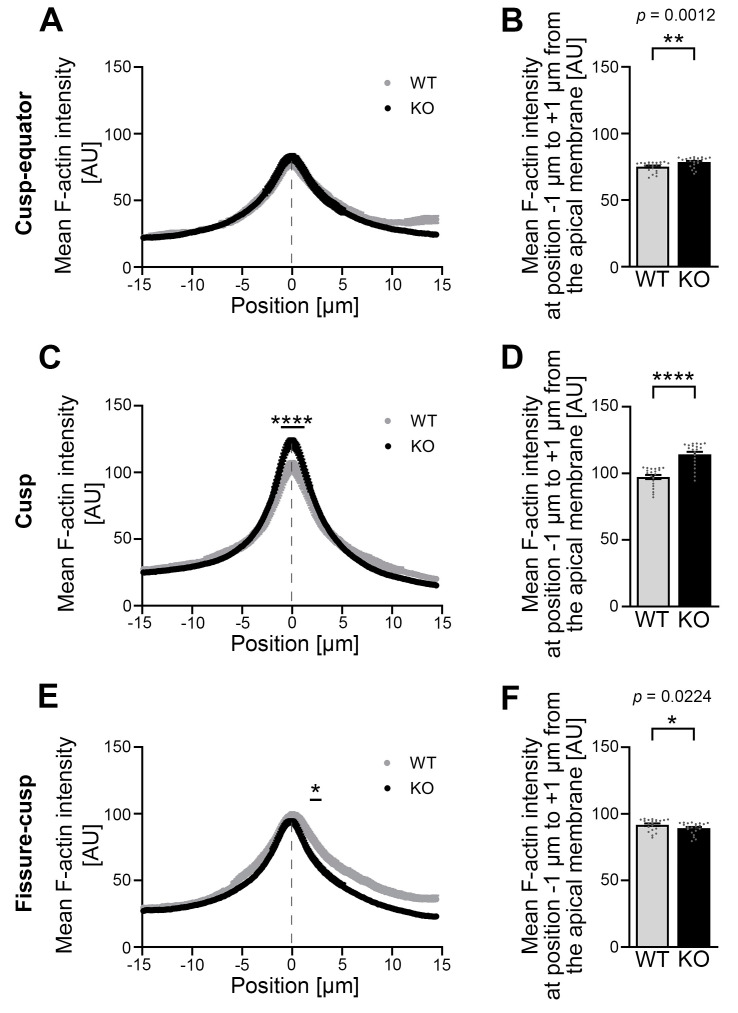
The mean F-actin intensity is increased in the apical membrane area of *Cobl* KO P0 ameloblasts, especially in the cusp area, compared to the WT. (**A**–**F**) The F-actin intensity was determined using phalloidin-stained cryosections in the apical membrane region of P0 ameloblasts of mandibular molars in the cusp–equator (**A**,**B**), cusp (**C**,**D**) and fissure–cusp (**E**,**F**) regions. Dashed lines mark the position of the plasma membrane. (**A**,**C**,**E**) show the spatially resolved mean F-actin intensity for WT and *Cobl* KO for 294 points (positions −15 μm to +15 μm) with the mean plasma membrane position in the center (position 0 μm). (**B**,**D**,**F**) represent the mean F-actin intensity in the area of the apical membrane (averaged measured values from positions −1 μm to +1 μm around the mean plasma membrane position (center; position 0 μm)). Note that the mean F-actin intensity in the cusp–equator region under Cobl deficiency was significantly increased compared to the WT, reflected in the statistical evaluation of the apical membrane region (−1 μm to +1 μm) (**B**). The mean F-actin intensity in the cusp region was highly significantly increased in *Cobl* KO compared to the WT (**C**,**D**). Only in the fissure–cusp region, Cobl deficiency in the apical, subcortical cytosol of the ameloblasts led to a significant reduction in mean F-actin intensity compared to WT (**E**). (**A**), WT, n = 56 (3 animals), *Cobl* KO: n = 122 (5 animals); (**C**), WT, n = 68 (3 animals), *Cobl* KO, n = 140 (5 animals); (**E**), WT, n = 69 (3 animals), *Cobl* KO, n = 140 (5 animals)). (**B**,**D**,**F**), WT, n = 21 (3 animals), *Cobl* KO: n = 21 (3 animals). Data mean ± SEM. Bar/dot plots (**B**,**D**,**F**). Two-way ANOVA with subsequent Šídák’s multiple comparisons tests (**A**,**C**,**E**) and Mann–Whitney tests (**B**,**D**,**F**). *, *p* < 0.05; **, *p* < 0.01; ****, *p* < 0.0001.

**Figure 6 cells-14-00359-f006:**
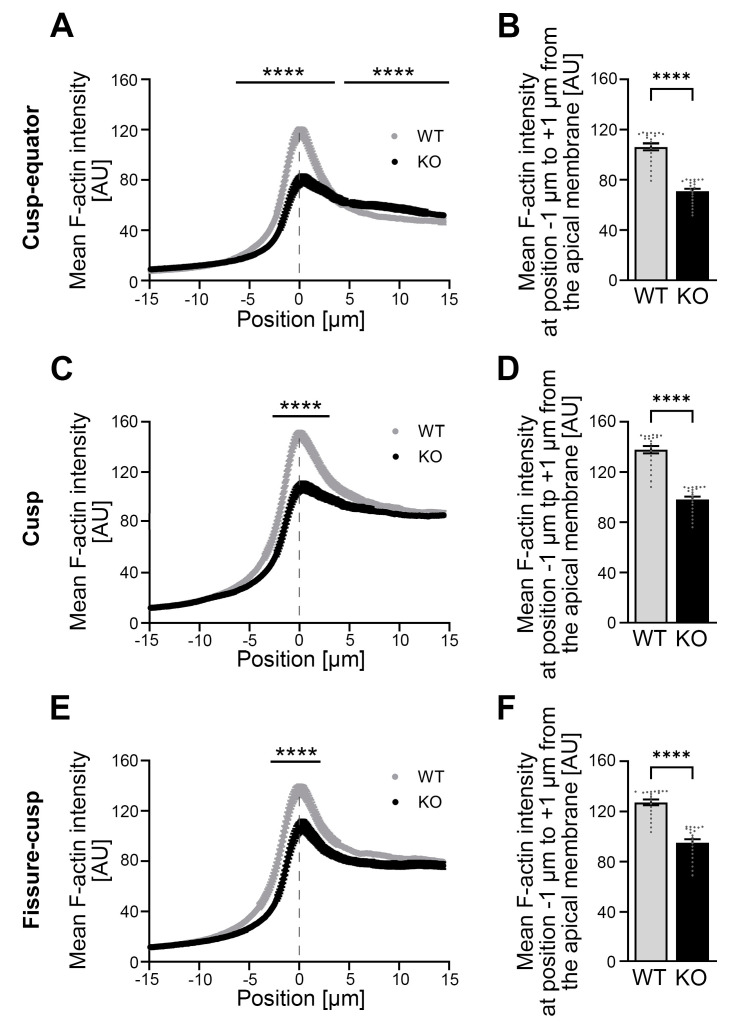
The mean F-actin intensity in the apical membrane area of P10 ameloblasts in *Cobl* KO in the cusp–equator, cusp and fissure–cusp regions is strongly reduced compared to the WT. (**A**–**F**) The F-actin intensity was determined in the apical membrane area of P10 ameloblasts of mandibular molars in the cusp–equator (**A**,**B**), cusp (**C**,**D**) and fissure–cusp (**E**,**F**) regions using phalloidin-stained cryosections. Dashed lines mark the mean position of the plasma membrane. (**A**,**C**,**E**) represent mean F-actin intensity spatially resolved for 294 points (−15 μm to +15 μm around the apical ameloblast plasma membrane) for WT and *Cobl* KO. (**B**,**D**,**F**) represent the mean F-actin intensity in the region of the apical membrane, whereby the averaged measured values around the center of the distance (−1 μm to +1 μm) were taken from the spatially resolved measurements (**A**,**C**,**E**). Note that the mean F-actin intensity in and around the apical membrane area was very strongly and highly statistically significantly reduced in *Cobl* KO compared to WT in all regions examined (**A**,**C**,**E**). For reasons of clarity, positions with low significance are not shown in (**A**,**C**,**E**). When considering the area of the apical membrane (−1 μm to +1 μm), a highly significant reduction in mean F-actin intensity under Cobl deficiency compared to the WT (**B**,**D**,**F**) was also observed in all regions examined. Only in the apical cytosol of the cusp–equator region the mean intensity of F-actin in *Cobl* KO was highly significantly increased compared to the WT (**A**). (**A**), WT: n = 80 (3 animals), *Cobl* KO: n = 120 (3 animals); (**C**), WT: n = 106 (3 animals), *Cobl* KO: n = 120 (3 animals); (**E**), WT: n = 98 (3 animals), *Cobl* KO: n = 110 (3 animals)). (**B**,**D**,**F**), WT: n = 21 (3 animals), *Cobl* KO: n = 21 (3 animals). Data mean ± SEM. Bar/dot plots (**B**,**D**,**F**). Two-way ANOVA with subsequent Šídák’s multiple comparisons tests (**A**,**C**,**E**) and Mann–Whitney tests (**B**,**D**,**F**). ****, *p* < 0.0001.

## Data Availability

The original contributions presented in this study are included in the article. Further inquiries can be directed to the corresponding authors.

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
