# Peer review of "The Evolutionary Young Actin Nucleator Cobl Is Important for Proper Amelogenesis"

_cells, 2025, doi:10.3390/cells14050359_

Round 1
Reviewer 1 Report
Comments and Suggestions for Authors
In this manuscript, KO of the actin nucleator, Cobl, was examined to infer impacts of Cobl on ameloblasts and amelogenesis in mouse. The paper if very well written, clear, and interpretations of results appear to be sound. The work seems to be very thorogh and technically sound, so I have little critique or requests for alteration of the text/figures. The authors might consider the following very minor points in revising/improving their manuscript:
1) line 99. Change ‘sacrification’ to ‘sacrifice’? Or is ‘sacrification’ commonly used?
2) line 466. Change ‘dring’ to ‘during’.
3) line 595. Change ‘uncoveres’ to ‘uncovers’.
4) line 684. Maybe add a few sentences here as to what you mean by saying Cobl is and ‘evolutionarily young nucleator’? That is, what is the evidence that Cobl is evolutionarily ‘young’, how young is young, etc.
5) last few paragraphs of the Discussion. Maybe some of the more speculative discussion points at the end of the paper could be trimmed back, but I leave this to the authors’ and editors’ discretions.
Reviewer 2 Report
Comments and Suggestions for Authors
This is a study designed to explore the importance of the actin nucleator Cordon-Bleu (Cobl) during formation of tooth enamel by ameloblasts. To date, this is the only study that has explored this area. The investigators use a combination of in situ hybridization to detect mRNA and immunofluorescence to examine protein expression and localization of Cobl in enamel development of mice. Cobl was found to be localized in a pattern very similarly to F-actin, with a particular enrichment in the apical plasma membrane of the ameloblasts. The importance of Cobl in enamel development was further explored using Cobl KO mice, by measuring the mean carbon content of the enamel. Cobl KO had a significantly increased mean carbon content compared to WT, which comes at the expense of the main elements of hydroxylapatite (calcium, oxygen and phosphate). Surprisingly, loss of Cobl caused an increased height of ameloblasts and an increased F-actin content at their apical membrane at the P0 stage of mouse development, while the F-actin density at the apical membrane was significantly reduced during the maturation phase of the enamel.
The experiments are well-done, the conclusions are supported by high quality data, and the figures are easy to understand and appropriate for the content. This work will be of broad interest to people in the field of actin biology.
I only have minor edits/suggestions:
- It would be interesting to look at the changes in other actin nucleators (Arp2/3 or members of the formin family) in the Cobl KO mice. Do they change their expression patterns in response to loss of Cobl? Do the authors think other actin nucleators can compensate for loss of Cobl?
- In addition to being an actin nucleator, Cobl has also been shown invitro to be able to sever actin filaments. Do the authors think that Cobl severing activity plays any role in enamel development?
- The authors state they use immunohistochemistry to observe Cobl expression (Line 333). I think they mean immunofluorescence.
Reviewer 3 Report
Comments and Suggestions for Authors
This manuscript deals with the expression and role of a particular actin nucleator protein Cordon blue or Cobl during the process of emanel formation in developing mandibular molar teeth. The authors analyse expression of Cobl in ameloblasts and its colocalisation with F-actin at their apical (enamel) and basal faces of molar teeth in wt and Cobl-ko mice during different phases of enamel secretion, deposition and maturation by immunostaining techniques. Comparison of Cobl wt and ko mice indicates that that Cobl is involved in mineral secretion and the transformation of ameloblasts from a columnar to cubic form. In addition, the F-actin content does not decrease to the same extent in Cobl-ko as in wt mice. Furthermore, the mineralization of pre-enamel is reduced in Cobl-ko mice.
In summary the authors present interesting data about a specialized developmental subject that are well suited for publication in Cells.
Minor points for amendment
1) Cordon blue is only mentioned in the key words, not explain within the main text.
2) A more detailed description of the special nature and activities of Cobl would be helpful. Why this nucleator and not one of the many other ones.
3) Are there other data suggesting that Cobl might act as an actin sequestering factor.
4) Do Cobl-ko mice finally produce non- or functionally impaired molar teeth?
5) Are there other gene defects in Cobl-ko mice as it appears that the mice were of general Cobl -/- genotype. Other organs were apparently analysed , but no results are mentioned.
